# A Comparative Transcriptome Analysis, Conserved Regulatory Elements and Associated Transcription Factors Related to Accumulation of Fusariotoxins in Grain of Rye (*Secale cereale* L.) Hybrids

**DOI:** 10.3390/ijms21197418

**Published:** 2020-10-08

**Authors:** Khalid Mahmood, Jihad Orabi, Peter Skov Kristensen, Pernille Sarup, Lise Nistrup Jørgensen, Ahmed Jahoor

**Affiliations:** 1Nordic Seed A/S: Grindsnabevej 25, 8300 Odder, Denmark; jior@nordicseed.com (J.O.); pskr@nordicseed.com (P.S.K.); pesa@nordicseed.com (P.S.); ahja@nordicseed.com (A.J.); 2Department of Agroecology, Faculty of Technology, Aarhus University, Forsøgsvej 1, Flakkebjerg, DK-4200 Slagelse, Denmark; lisen.jorgensen@agro.au.dk; 3Department of Plant Breeding, The Swedish University of Agricultural Sciences, 23053 Alnarp, Sweden

**Keywords:** *cis*-regulatory elements, disease resistance, Fusarium head blight, GO enrichment, hybrids, transcriptome, rye

## Abstract

Detoxification of fusariotoxin is a type V Fusarium head blight (FHB) resistance and is considered a component of type II resistance, which is related to the spread of infection within spikes. Understanding this type of resistance is vital for FHB resistance, but to date, nothing is known about candidate genes that confer this resistance in rye due to scarce genomic resources. In this study, we generated a transcriptomic resource. The molecular response was mined through a comprehensive transcriptomic analysis of two rye hybrids differing in the build-up of fusariotoxin contents in grain upon pathogen infection. Gene mining identified candidate genes and pathways contributing to the detoxification of fusariotoxins in rye. Moreover, we found *cis* regulatory elements in the promoters of identified genes and linked them to transcription factors. In the fusariotoxin analysis, we found that grain from the Nordic seed rye hybrid “Helltop” accumulated 4 times higher concentrations of deoxynivalenol (DON), 9 times higher nivalenol (NIV), and 28 times higher of zearalenone (ZEN) than that of the hybrid “DH372” after artificial inoculation under field conditions. In the transcriptome analysis, we identified 6675 and 5151 differentially expressed genes (DEGs) in DH372 and Helltop, respectively, compared to non-inoculated control plants. A Kyoto Encyclopedia of Genes and Genomes (KEGG) analysis revealed that DEGs were associated with glycolysis and the mechanistic target of rapamycin (mTOR) signaling pathway in Helltop, whereas carbon fixation in photosynthesis organisms were represented in DH372. The gene ontology (GO) enrichment and gene set enrichment analysis (GSEA) of DEGs lead to identification of the metabolic and biosynthetic processes of peptides and amides in DH372, whereas photosynthesis, negative regulation of catalytic activity, and protein-chromophore linkage were the significant pathways in Helltop. In the process of gene mining, we found four genes that were known to be involved in FHB resistance in wheat and that were differentially expressed after infection only in DH372 but not in Helltop. Based on our results, we assume that DH372 employed a specific response to pathogen infection that led to detoxification of fusariotoxin and prevented their accumulation in grain. Our results indicate that DH372 might resist the accumulation of fusariotoxin through activation of the glycolysis and drug metabolism via cytochrome P450. The identified genes in DH372 might be regulated by the WRKY family transcription factors as associated *cis* regulatory elements found in the in silico analysis. The results of this study will help rye breeders to develop strategies against type V FHB.

## 1. Introduction

Fusarium head blight (FHB) is a devastating disease of cereals and is caused by trichothecene-producing pathogens belong to *Fusarium* spp. [1,2]. The species *F. graminearum* and *F. culmorum* are regarded as the two most important *Fusarium* species globally, which infect all members of the *Gramineae/Poaceae* family [3,4]. The cereals are susceptible to infection from the flowering stage (anthesis); hence, early FHB infections start around anthesis, leading to floret sterility or poor grain filling and subsequently resulting in huge yield losses [5]. In addition, FHB led to accumulation of high content of mycotoxins that contaminate the grain and, when ingested, can adversely affect livestock and human health [6,7]. Due to their acute toxicity to human and animals, several countries have put strict regulation on the permissible level of fusariotoxins in grains. Moreover, fusariotoxins such as deoxynivalenol act as a virulence factor for Fusarium, facilitating disease spread within spikes [8]. The resistance to FHB is a complex mechanism, and various components of resistance have been reported in the literature [9]. Based on mechanisms of resistance, five types of FHB resistance are recognized [10,11]. Resistance to initial infection (type I) and to the spread of infection within spikes (type II) remain the two most widely accepted forms of resistance [12]. The remaining three types of FHB resistances include the resistance to kernel infection (type III), host tolerance (type IV), and resistance to accumulation of mycotoxin or fusariotoxins (type V) [11,13]. Type V resistance is commonly considered a component of type II resistance, as it typically reduces spread of disease [8]. Type V resistance is further subdivided into two classes, i.e., class 1: processes that chemically modify mycotoxin to a less toxic form and class 2: processes that prevent the accumulation of mycotoxins [13]. The fusariotoxins that contaminate the grain typically consist of deoxynivalenol (DON), nivalenol (NIV), and zearalenone (ZEN) [14]. The most widely reported form of detoxification of DON is by UDP-glucosyltransferase (UGT), which glucosylates DON to the less toxic DON-3-O-glucoside (D3G) [15]. To date, we have very limited understanding of the detoxification mechanisms of FHB due to complexities such as synthesis of various mycotoxins, chemotypes, and numerous infection mechanisms. In order to overcome the FHB, a better understanding of detoxification mechanism (type V resistance) and dissection of transcriptional landscape during host–pathogen interaction is needed. A number of genes such as glycosyltransferases, ATP-binding cassette transporters (ABC) transporters, and cytochrome P450s have been known to contribute FHB resistance through detoxification of fusariotoxins [16].

The genetics of FHB resistance have been investigated in bread wheat [17,18], durum wheat [19], barley [20], and to some extent in triticale [21], but less efforts have been made in rye. Rye (*Secale cereale* L.) is an allogamous, diploid (RR, 2*n* = 2x = 14), and widely grown cereal crop in northern Europe [22]. In recent years, it has also been used for substrate for bioethanol and biogas production [23,24]. Rye is reported to have a large genome of approximately 7.9 Gb [22]. The genetics of FHB resistance are complex since this trait is under multigenic control [25,26,27,28], and substantial genotypic variation to FHB resistance in breeding populations of rye has been reported [29,30,31]. In wheat, several hundred quantitative trait loci (QTLs) linked to FHB resistance were mapped [26,32,33,34,35]. In fact, a major QTL in Chinese spring wheat (Sumai 3) provides the most stable and the largest effect on FHB resistance in wheat [36]. Despite the overwhelming number of known QTLs, there is scarcity of information about candidate genes. Marker-assisted breeding of the majority of these QTLs results in transfer of undesirable traits due to linkage drag [37,38,39]. Hence, the identification of candidate genes linked to a specific type of FHB resistance could be useful for breeding resistant cultivars without compromising other desirable traits. Recently, a gene that encodes a putative histidine-rich calcium-binding protein (TaHRC) was identified in wheat as *Fhb1* and a deletion spanning the start codon of this gene resulted in FHB resistance [40]. Due to the abovementioned complexities associated to FHB, the determination of the exact number of genes responsible for resistance to FHB is very limited and obtaining that information can be a formidable task. However, the integration of various omics platforms can yield fruitful results. In fact, integration of QTLome with genomic and transcriptomic data has been exploited successfully for the discovery of promising candidate genes linked to FHB resistance [21,41,42,43,44]. Using this approach, 10 promising candidate genes have been identified in wheat including the TaHRC, glycosyltransferses, and cytochrome P450 [41]. However, unlike other cereal crops affected by FHB, very little omics resources are available for rye. Until now, only a single genome-wide association study (GWAS) has been reported in rye for FHB resistance [14]. The transcriptomics data generated in this study will be a valuable asset to enhance overall omics resources for rye, particularly keeping in view that there is not a single study in rye aiming to decipher FHB resistance through utilizing transcriptomic profiling.

RNA sequencing combined with well-developed bioinformatic tools can identify candidate genes and can find the functional elements that are active against spread of FHB. Therefore, the aim of this study was to characterize the transcriptome of rye after Fusarium infection. The rye hybrids included in this study accumulated contrasting levels of fusariotoxins after pathogen inoculation. Therefore, they provided a good experimental setup to decipher the detoxification mechanism of fusariotoxins. As detoxification mechanisms is associated to type V FHB resistance, the comparison of transcriptome between two hybrids also shed light on this specific type of resistance. In short, we have undertaken the first global analysis of rye transcriptome using two rye hybrids having contrasting levels of fusariotoxins after inoculation with *Fusarium* spp. The differential expression of disease-responsive genes was profiled, and the molecular pathways involved were analyzed. Moreover, a list of differentially expressed genes (DEG) including the promising candidate genes responsible for detoxification of fusariotoxins in rye was developed. To get an in-depth understanding about the differentially expressed genes, we identified conserved regulatory elements and associated them to specific transcription factors (TFs). These motifs and associated TFs might be responsible for controlling the expression of genes that detoxify fusariotoxins. Furthermore, in this study, we have implemented various bioinformatics tools and analyzed the transcriptome data of independent experiments deposited in global databases. This is the first study on expression profiling of FHB-related genes in rye hybrids using next generation sequencing (NGS) technology. It is expected that this work will serve as a valuable scientific resource for better understanding of a specific type of FHB resistance (type V) in rye.

## 2. Results

### 2.1. Disease Severity and Mycotoxin Analysis

The assessment of disease severity was conducted on both hybrids after artificial inoculation under field conditions. The disease symptoms were not significantly different between the Helltop and DH372 hybrids (Table 1). However, mycotoxin analysis of the grains revealed significant differences among the mycotoxin content in both hybrids (Table 1). In the mycotoxin analysis, we quantified the deoxynivalenol (DON), the nivalenol (NIV), and the zearalenone (ZEN) toxin in both hybrids. The fusariotoxins concentrations were very high in the grains of Helltop. Helltop grains accumulated 4 times higher concentration of DON, 9 times higher NIV, and 28 times higher of ZEN than that of DH372 after artificial inoculation under field conditions (Table 1).

### 2.2. Differential Gene Expression Analysis in Different Treatments

Twelve samples of RNAseq reads, each containing three independent biological replicates of spikes of Helltop and DH372 with or without infection of *Fusarium* spp., were included in this investigation (Figure 1). In the cases of DH372 and Helltop, we identified 6675 and 5151 DEGs, respectively (Table 2). Of these DEGs, 3144 and 2688 were downregulated in DH372 and Helltop compared to uninfected, respectively. Similarly, 3531 and 2463 DEGs were upregulated in DH372 and Helltop compared to non-inoculated control plants, respectively (Table 2). Furthermore, the comparison of identified DEGs in Helltop and DH372 were presented in the form of a Venn diagram (Figure 2A). In this comparison, 1794 DEGs were found in both hybrids, whereas 4881 and 3357 genes were differentially expressed distinctively in only DH372 or Helltop, respectively (Figure 2A). A total of 4850 DEGs was found in the comparison made between *Fusarium* inoculated DH372 and *Fusarium* inoculated Helltop, with 2623 downregulated and 2227 upregulated in DH372. Similarly, DEGs in the control samples of Helltop and DH372 (without infection), we identified 116 DEGs. Among these DEGs, 72 were downregulated and 44 upregulated in DH372 compared to Helltop (Table 2).

### 2.3. Enzyme Classes, KEGG, and Gene Enrichment Analysis of DEGs

Due to the high number of DEGs, we explored the function of these genes through assigning them to various enzyme classes, using the Kyoto Encyclopedia of Genes and Genomes (KEGG) pathways and gene enrichment analysis. For this purpose, we divided these DEGs into three categories, as presented in a Venn diagram (Figure 2A), and named them exclusive DH372 (DEGs found only in DH372), exclusive Helltop (DEGs found only in Helltop), and common (DEGs found in both hybrids). The DEGs in exclusive DH372, exclusive Helltop, and the common category have 809, 754, and 324 genes, respectively, that have enzyme codes (Figure 2B). The shares of these enzyme classes were 17%, 22%, and 17% of all DEGs in exclusive DH372, exclusive Helltop, and the common category, respectively (Figure 2B). A high number of these DEGs were associated with hydrolases, followed by transferases and oxidoreductases in all three categories. The KEGG analysis was used to identify potential pathways represented in the DEGs. The top twelve KEGG pathways, based on the number of genes identified in our results, were compared among DEGs of the abovementioned categories. The most represented pathways are important basic pathways necessary for cell life and, therefore, naturally found in DEGs. In metabolism pathways, the DEGs exclusive to DH372 has higher representation in purine, thymine, and drug metabolism via cytochrome P450, whereas Helltop has a higher number of DEGs in starch and sucrose metabolism, amino sugar nucleotide, sugar metabolism, and drug metabolism via other enzymes (Figure 2C). The DEGs associated to glycolysis and the mTOR signaling pathway were higher in DH372, whereas carbon fixation in photosynthesis organisms, and pentose and glucuronate interconversions were higher in Helltop compared to DH372.

After enzyme classes and KEGG pathway analysis, we used these DEGs of three categories for gene enrichment analysis. Here, we conducted gene enrichment separately for upregulated and downregulated DEGs for deeper insights. The top 15 genes enrichment terms were presented for each category (Figure 3 and Figure 4). In case of upregulated DEGs in DH372, the significantly enriched gene ontology (GO) terms were associated with integral component of membrane, ATP binding, nucleobase-containing compound biosynthetic processes, carbohydrate metabolic process, kinase activity, and transmembrane transport activity (Figure 3A). Similarly, in the case of upregulated DEGs in Helltop, the significantly enriched GO terms were an integral component of the membrane, oxidoreductase activity, transmembrane transport activity, hydrolase activity, cell wall, and photosystem (Figure 3B). The significantly enriched terms in common DEGs were integral components of membrane, ATP binding, oxidation–reduction processes, kinase, phosphorylation, and hydrolase activity (Figure 3C). Overall, these results revealed that all three categories in upregulated DEGs have few common enriched terms such as integral component of membrane, whereas substantial differences were observed in the significant gene-enriched GO terms of each category. The significant enriched GO terms in common DEGs are more similar to Helltop than the enriched terms of DH372 (Figure 3). We had observed similar trends in the downregulated DEGs as the gene-enriched terms of common DEGs are more similar to Helltop than that of DH372 (Figure 4).

After GO term enrichment analysis, we decided to use gene set enrichment analysis (GSEA) to identify the most significant biological processes in the DEGs of Helltop and DH372 after pathogen infection. (Figure 5). This analysis led us to identify the metabolic and biosynthetic processes of peptides and amides in DH372 as the most significant biological processes, whereas photosynthesis, negative regulation of catalytic activity, and protein-chromophore linkage were the significant pathways in Helltop. Both GO term enrichment and GSEA revealed the different distinctive responses of DH372 and Helltop in the case of pathogen infection.

### 2.4. Dissection of DEGs for Mining of Candidate Genes

The results of KEGG pathways and enrichment analysis prompted us to do further dissections of differentially expressed genes to get potential candidate genes involved in detoxification of fusariotoxin in DH372 and resistance to *Fusarium* spp. We compared the DEGs and divided them into three different categories (Figure 6). These categories include exclusively responding genes, common responding genes, and contrasting responding genes categories. Here, the exclusively responsive genes are the ones that were differentially expressed in one hybrid after inoculation but not in the contrasting hybrid. The common responding genes are the ones that exhibited similar differential expression patterns in both hybrids, whereas contrasting responding genes are the one that exhibited opposite responses in one hybrid compared to the other after pathogen infection. Among the exclusively responding genes, 2562 and 2319 DEGs were exclusively up- and downregulated in DH372, whereas 1763 and 1594 genes were exclusively up- and downregulated in Helltop (Figure 6A). Among common responding genes, 666 DEGs were up- and 791 were downregulated in both hybrids (Figure 6A). Interestingly, 337 genes were identified as contrasting responding genes after pathogen infection. Among these contrasting responding genes, 303 were upregulated in DH372 while the same candidate genes exhibited opposite response in Helltop (Figure 6A). Similarly, in contrasting responding genes, 34 DEGs were upregulated in Helltop, while these were downregulated in DH372 (Figure 6A). In the fusariotoxin analysis, we found significant difference in the contents of DON, NIV, and ZEN; hence, in order to reduce complexity, we focused on gene families responsible for detoxification of mycotoxin. These gene families include candidate genes known as glycosyltransferases (GT) and cytochrome P450s and hence checked their behavior after pathogen infection in both hybrids. Another reason to choose GT and cytochrome P450s is that these genes predominantly constitute the pathways and biological processes identified in our KEGG and gene enrichment analysis. Among the DEGs belonging to cytochrome P450, four genes were part of the common responding genes category, whereas 25 cytochrome P450 genes were identified in the exclusively responding genes category (Figure 6B). Among the common responding category of cytochrome P450, two genes were upregulated and two were downregulated in both hybrids after inoculation with *Fusarium* spp. (Figure 6B). The upregulated cytochrome P450 genes were named “XLOC_114701” and “XLOC_996749” with four-fold change in DH372 after inoculation, whereas in Helltop, these were upregulated significantly with 12- and 3-fold changes, respectively (Table 3). The downregulated cytochrome P450 genes were named “XLOC_939724” and “XLOC_180117” with minus 284- and 27-fold changes in DH372, whereas in Helltop, these were downregulated with 473 and 3 fold changes, respectively (Table 3). In the case of GT, only three GTs fall in the common responding category and all three were downregulated in both hybrids, whereas 25 GTs were identified in the exclusively responding category (Figure 6C). The GTs in the common responding category were named “XLOC_1029148”, “XLOC_128937”, and “XLOC_1336988” and were downregulated in both hybrids (Table 3). In the case of the contrasting responding category, we identified neither any cytochrome P450 nor any GT. Interestingly, four genes in the exclusive responding category of DH372 are also identified in wheat and involved in FHB resistance [41]. Among them, three were identified as P450s and glycosyltransferase. These genes were named “XLOC_1213553”, “XLOC_165003”, and “XLOC_1267741” with upregulation at 7-, 5-, and 2-fold changes, respectively (Table 4). The fourth one was named “XLOC_181573” with a 4-fold change in downregulation and was annotated as a metal tolerance protein (Table 4). These genes were differentially expressed only in DH372 but not in Helltop after pathogen infection. In order to validate the expression of these identified genes in both categories, we decided to explore the orthologs of these genes in *T. aestivum* using publicly available gene expression databases. 

### 2.5. Gene Expression Profiling of Orthologs of Specific GT and Cytochrome P450s Identified in Common Responding and Exclusive Responding Category

The orthologs of seven candidate genes (GT and cytochrome P450s) in the common responding category were extracted from reference genome of *T. aestivum* using sequence similarity searches (protein evidence). In order to check the expression of these genes belonging to cytochrome P450s and glycosyltransferases, we utilized the Genevestigator software to assess their behavior under various stresses. During different stages of growth, orthologs of these genes in wheat had very similar expression patterns to each other with the highest expression during flowering and heading stage (Figure 7A). The cytochrome P450s (TraesCS1A02G351600, TraesCSU02G008900, TraesCS6B02G205100, and TraesCS6D02G193600) had similar expression patterns throughout the developmental stages with maximum expression during flowering and heading stage (Figure 7A). Similarly, GTs (TraesCS4A02G401800, TraesCS4B02G322500, and TraesCS1D02G005200) are highly expressed throughout all developmental stages and tend to have decreased expression at senescence (Figure 7A). Through using the condition tool of the Genevestigator, we found out that the *Fusarium* infection causes significant changes in their expression (*p* < 0.001 and fold change > 3.00) and that the trend is very similar to what we observed in our study (Figure 7B). This shows the conserved functions of these genes in wheat and rye during *Fusarium* infection. Similarly, the orthologs of identified glycosyltransferases (TraesCS4A02G401800, TraesCS4B02G322500, and TraesCS1D02G005200) in the common responsive category were also found to be significantly downregulated after *Fusarium* infection (Figure 7B). Interestingly, all seven genes in the common responding category has exactly the similar behavior as that found in our study in rye after *Fusarium* inoculation in both hybrids. In the quest to understand the function of the four genes identified in the exclusive responding category, we found that all four genes in wheat were highly expressed during the flowering stage with maximum expression at various developmental stages (Figure 7C). The cytochrome P450 (TraesCS3B02G026200) and one glycosyltransferase (TraesCS3B02G021700) had the highest expressions at flowering, whereas second glycosyltransferase (TraesCS3B02G021100) and metal tolerance protein (TraesCS3B02G040900) had the highest expressions at the tillering and senescence stages, respectively. The in silico analysis during various stresses revealed that these genes were differentially expressed during biotic and abiotic stresses and nutrient stimulus. From these genes, three were differentially expressed after *Fusarium* infection (Figure 7D). These are also similar to our findings in rye except for cytochrome P450 (TraesCS3B02G026200), which is significantly downregulated in wheat. Nonetheless, three out of four were differentially expressed in this analysis after pathogen inoculation.

### 2.6. Cis Motif Identification and Associated Transcription Factors

We analyzed the promoter regions of the identified genes and reported the top five common motifs. The motifs found in promoters of selected genes in the common responsive category are presented in Table 5, whereas the motifs of selected DEGs exclusively found in DH372 are presented in Table 6. In a Gene Ontology for Motifs (GOMO) analysis, three motifs in the common responsive category and two in the exclusive responsive category were found to be involved in transcription factor activity (Table 5 and Table 6). In order to find the role of motifs associated to the transcription factor activity, the PlantPAN promoter analysis tool was utilized to explore TF binding site with a cutoff value (*q*-value < 0.05). In the common responding category, motif 1 was found to be linked to MYB-like transcription factors, whereas motif 2 was found to be linked with WRKY family transcription factors. We identified six MYB family transcription factors that were differentially expressed in both hybrids after inoculation (Table 7). Similarly, we found that three WRKY family transcription factors were differentially expressed only in DH372 but not in Helltop after pathogen infection (Table 8). 

## 3. Discussion

The detoxification of fusariotoxin is a type V of FHB resistance and considered as a component of type II resistance, which is related to spread of infection within spikes [11]. Understanding this type of resistance is vital for FHB resistance, but to date, nothing is known about candidate genes that confer this type of resistance in rye as we lack genomic resources. In this study, we generated transcriptomic resources, identified DEGs and *cis* regulatory elements, and linked them to transcription factors, which might regulate expression of candidate genes responsible for detoxification of fusariotoxins. The fusariotoxins analysis in this study revealed that two rye hybrids have different detoxification abilities as one (Helltop) accumulated a higher level of fusariotoxins than the other (DH372) (Table 1). Furthermore, transcriptome data generated in this study complements the resources recently made available for other cereals regarding FHB resistance [43,45]. Wheat and barley differ noticeably in this type of resistance; however, there is not much known in the case of rye. Wheat typically possesses this type of resistance, whilst in contrast, barley is generally highly resistant to fungal spread [46]. Mycotoxin does not appear to possess a role of virulence factor during infection of barley heads [47]. Based on these results, we assumed that rye is more similar to barley as we did not observe differences in the visual symptoms of diseases even though the level of fusariotoxin was significantly different between two hybrids after pathogen inoculation. This could also be due to fact that we might need more sensitive tools than visual scoring in the case of rye. For example, researchers found that, in the case of resistant lines in wheat, evaluation of FHB resistance based on sensitive tools such as real time PCR is more reliable than visual scoring [48]. Rye is more resistant to diseases than wheat; hence, sensitive tools for evaluation of disease would be more appropriate here.

Plant resistance to *Fusarium* and production of various fusariotoxins are a highly complex mechanism [49]. Different levels of fusariotoxins in the mycotoxin analysis chiefly prompted us to perform a transcriptomic study to get deeper insights. Generally, in the transcriptomic study, we identified higher numbers of DEGs after inoculation in both hybrids. These results are in agreement with previous studies involving wheat, where plants were challenged with *Fusarium* species [45,50,51]. The KEGG analysis on DEGs revealed that DEGs associated with glycolysis and the mTOR signaling pathway were higher in the hybrid having less fusariotoxins (DH372), whereas carbon fixation in photosynthesis organisms, and pentose and glucuronate interconversions were most represented in the hybrid having higher levels of fusariotoxins (Helltop). These annotations of DEGs can serve important clues for the understanding of early response of both rye hybrids against this pathogen. The differences found in drug metabolism via cytochrome P450 and other drug metabolism enzymes, mTOR signaling pathway, pentose and glucuronate interconversions could be the reason for differences observed in mycotoxin analysis of both hybrids. Similar KEGG pathways were identified in wheat [45,52,53], barley [54], and maize [55] after *Fusarium* infection. The glycolysis pathways have been known to play significant roles in plant pathogen interaction [56]. Drug metabolism constitutes the genes belonging to family of cytochrome P450s, GSTs, glycosyl transferases, and ABC transporters. These pathogenesis-related genes are known to play a role in plant–pathogen interactions in various plant species. Hence, this agrees with the notion that plant defense depends on metabolism of pathogenesis-related proteins at the pathogen contact site [57,58,59,60]. Another KEGG pathway was pentose and glucoronate interconversions that can increase the strength of cell walls, eventually leading to resistance to pathogens [61]. Overall, the KEGG analysis identified the recruitment of different pathways in response to *Fusarium* infection that could explain differences in the contents of mycotoxin between Helltop and DH372. These results encouraged us to do a GO enrichment analysis for upregulated and downregulated DEGs separately for deeper insights. The significant enriched terms in common DEGs are more similar to Helltop than enriched terms of DH372 (Figure 3 and Figure 4). Based on GO enrichment analysis, we can speculate that DH372 employed an additional response that constitutes the detoxification of fusariotoxins after pathogen infection. The GSEA analysis unveiled completely unrelated biological processes using the DEGs of DH372 and Helltop. In the case of DH372, we identified the metabolic and biosynthetic processes of peptides and amides, whereas photosynthesis, negative regulation of catalytic activity, and protein–chromophore linkage were the significant pathways in Helltop (Figure 5). The metabolic process of peptide and amides constitute integral parts of the plant immune system and can have a profound impact on plant resistance to specific pathogen types [62]. It has been known that FHB infection affects photosynthetic activity in plants [63]. These different pathways upon pathogen infection might exhibit differences in the costs of FHB resistance in these hybrids after inoculation. Nevertheless, these results indicate the presence of different responses of the hybrids upon pathogens at the molecular level.

Since a large number of genes constitute these biological pathways, we decided to do gene mining and focused on specific pathways known to play significant roles in detoxification of fusariotoxins. In the process of gene mining, we focused on the glycolysis and drug metabolism via P450s as these pathways are known to be responsible for detoxification of mycotoxin. Moreover, glycosyltransferases (GTs) are the principal agents by which host plants transform DON into the less toxic form deoxynivalenol-3-O-glucoside [15]. A number of GTs have been identified in cereal crops that contribute to improving the disease resistance to FHB [64,65,66]. Similarly, cytochrome P450s are associated with FHB resistance in several earlier studies [44,67,68]. The meta-analysis of the QTLome of FHB resistance by Venske et al., 2019, identified 10 candidate genes responsible for FHB resistance in bread wheat [41]. Among them, three belong to GTs and cytochrome P450. In transcriptome profiling, we found the same four genes identified by Venske et al. only in DH372 that were differentially expressed but not in Helltop (Table 4). This is very interesting and might explain the lower level of accumulation of mycotoxin in the grains of DH372 after pathogen infection.

We characterized cytochrome P450s and GTs using the wheat orthologs in the transcriptome data deposited in publicly available databases. The selected GTs and cytochrome P450s belong to the common responding and exclusive responding categories. The seven-candidate genes in the common responding category and four candidate genes that were exclusively found in DH372. The four exclusive responding genes are the ones that were also identified in wheat and responsible for FHB resistance. The publicly available transcriptome data revealed that *Fusarium* infection causes significant changes in the expression of seven P450s and GTs, and this trend is very similar to what we observed in rye. These seven genes were differentially expressed in both hybrids upon pathogen infection. This indicates that these seven genes behave very similar in response to *Fusarium* infection and might constitute the conserved pathways in both species (rye and wheat) in response to *Fusarium* infection. Hybrid DH372 and Helltop had different levels of mycotoxin; therefore, GTs and cytochrome P450s belonging to common responding category cannot be components of the detoxification pathway. Hence, the four candidate genes belong to GT and cytochrome P450s and found only in DH372 can constitute the detoxification pathway. Using this notion, we can elucidate that DH372 has lower accumulation of mycotoxin in grains as it can detoxify fusariotoxin efficiently. These assumptions encouraged us to look for regulatory *cis* motifs for seven common responding genes and four genes exclusively found in DH372.

We had listed the 10 motifs belonging to selected genes of the common responding category and exclusive responding category. Among them, five motifs were found to be linked with transcription factors, whereas the remaining five motifs were not found to be linked to any biological process in the in silico analysis. Motif 2 of the exclusive responding category was found to be linked to WRKY family transcription factors. The WRKYs are one of the largest families of transcriptional regulators in plants and are involved in biotic and abiotic stress responses [69]. The WRKY transcription factors have been known in the regulation of transcriptional reprogramming associated with plant immune responses [70,71]. The constitutive overexpression of WRKY conferred an enhanced resistance against *F. graminearum* in transgenic wheat plants grown under greenhouse conditions [72]. The TaWRKY70 from wheat acts as a positive regulator of resistance against *F. graminearum* [73]. The expression of *WRKY29* is associated with pattern-triggered immunity (PTI) and is induced in response to *F. graminearum* infection [74]. In our transcriptome study, we identified that three WRKY family transcription factors were differentially expressed only in DH372 but not in Helltop after pathogen infection (Table 8). In the common responding category, motif 1 was found to be linked to MYB-like transcription factors, whereas the remaining motifs were not found to be associated significantly with any other transcription factors. We identified six MYB family transcription factors that were differentially expressed in both hybrids after inoculation (Table 7). MYB proteins are key factors in regulatory networks controlling development, metabolism, and responses to biotic and abiotic stresses [75]. These MYB proteins might regulate the pathways that represent common pathways in both hybrids. Overall, gene expression profile and *cis* regulatory elements could facilitate the understanding of type V of FHB resistance in rye at the molecular level.

## 4. Conclusions

Our transcriptome data provided comprehensive insights into gene expression profiles and facilitated a molecular mechanism study of Fusarium pathogen response in rye. A putative gene network underlying the molecular response in two rye hybrids with differential accumulation of mycotoxin was proposed. The transcriptomics data generated in this study will be a valuable asset to enhance overall omics resources for rye, particularly keeping in view that, previously, there was not a single study in rye aiming to decipher FHB resistance through utilizing transcriptomic profiling. The differential expression of disease-responsive genes was profiled, and the molecular pathways involved in both hybrids were analyzed. Based on results and different bioinformatic analysis of the identified DEGs, we assume that DH372 employed an additional response to pathogen infection that led to detoxification of mycotoxins and prevented their accumulation in grain. DH372 might resist the spread of *Fusarium* disease in grains mainly through the activation of glycolysis and drug metabolism via cytochrome P450, which were regulated by WRKY family transcription factors. The four candidate genes involved in detoxification of fusariotoxins were found only in DH372. This could be the reason that DH372 has lower accumulation of fusariotoxins in grains. Hence, these genes remain interesting candidate genes for future studies such as Clustered Regularly Interspaced Short Palindromic Repeats CRISPR to validate their function in a specific type of FHB resistance. Overall, gene expression profile and *cis* regulatory elements could assist the understanding of type V FHB resistance in rye at the molecular level. Moreover, availability of transcriptome sets from rye hybrids under *Fusarium* stress as presented here will be appreciated as a resource for genomics studies in rye.

## 5. Material Methods

### 5.1. Rye Hybrids, Inoculation, Field Disease Scoring, and Mycotoxin Analysis

The seeds of rye hybrids Helltop and DH372 were obtained from breeding company Nordic Seed A/S, Germany. Both hybrids consist of similar genetic components except the restorer component. The field trial was established in 2019 and located at Flakkebjerg, Denmark (Aarhus University field station). Both rye hybrids were tested for their response to *Fusarium* spp. under optimal infection conditions. Hybrids were grown in randomized manner in two replications with one row per m^2^ plots. Two *Fusarium* isolates, *F. graminearum* (strain 7775) and *F. culmorum* (strain 8984), were used to inoculate Helltop and DH372. Both strains (*F. culmorum* and *F.graminearum*) used in this study are characterized as DON producers. In previous studies, it has been found that these strains are common in Denmark [76]. Spore suspensions of both strains were prepared and sprayed according to Etzerodt et al. [77]. Briefly, spikes were sprayed directly with a *Fusarium* spp. macro-conidial suspension (2 × 10^6^ spores/mL) three times during anthesis using the knapsack sprayer (BBCH 60, 65, and 69) [78]. Inoculations took place in the evening to achieve optimal conditions for infection due to high humidity conditions. The disease severity of FHB was assessed at growth stage BBCH 85. A visual linear scale from 0 to 10 was used to assess the infected plants within each plot. The fusariotoxin accumulations (DON, NIV, and ZEN) in the seeds of inoculated hybrids were quantified in an LC-MS/MS system consisting of an Agilent 1260 infinity chromatographic system (Santa Clara, CA, USA) coupled to an AB Sciex 3200 triple quadrupole trap mass spectrometer (QTRAP/MS) (AB Sciex, Framingham, MA, USA), as described by Etzerodt et al. [77]. Analyst 1.6.3 software (AB Sciex, Woodlands Central Indus. Estate, Singapore), Framingham, USA) was used to control the LC-MS/MS system. The same number of seeds per hybrid was grinded and used for analysis of mycotoxins.

### 5.2. Plant Growth Conditions and Experimental Setup for Transcriptome Analysis

An experiment to obtain plant material for comparative transcriptome was conducted in environment-controlled growth chambers. Seeds of these two hybrids were sown in peat pots (diameter, 15 cm) in a greenhouse. After two weeks of germination, plants were placed in a cold room for vernalization for 8 weeks. The conditions were maintained at 4 °C, 8/16 h day/night cycle, 120 μE m^−2^ s^−1^ light intensity, and 80% humidity during vernalization.

After vernalization, plants were moved into two different growth chambers to avoid cross contamination designated as the control and were treated and inoculated with a spore solution of *Fusarium* spp. Both chambers were set at same growth conditions, i.e., 16 h light (8000 Lux; 18 °C) and 8 h darkness (15 °C). During growth, nutrients were supplemented once a week during watering. At flowering initiation stage, individual spikes were inoculated with either sterilized water (mock control) or with freshly prepared spore suspension of *Fusarium* spp., as described in the section above. A spore concentration at 1 × 10^6^ conidia/mL was used for inoculation. After spray, both treatments were covered with plastic bags to maintain humidity and to avoid cross contamination. Spikes were collected in triplicate after 5 days of post inoculation and immediately frozen in liquid nitrogen and stored at −80 °C until RNA extraction.

### 5.3. RNA Extraction and Sequencing

Total RNA was extracted from 50 mg from each of twelve individual spikes using the Cetyl trimethylammonium bromide (CTAB) method [79] with slight modifications. Briefly, spikes were grounded separately using Geno/Grinder (SPEX Sample prep. Stanmore. UK) for 45 sec at 1500hertz. The fine tissue powder was suspended in 1 mL of preheated (65 °C) CTAB extraction and 2% β-mercaptoethanol and was mixed carefully. The suspension was incubated in thermomixer at 65 °C for 15 minutes to permit the complete dissociation of nucleoprotein complexes and was centrifuged at 12,000× *g* for 5 min at 4 °C. Later, the supernatant was transferred to a new 2-mL tube and was added 400 μL of chloroform per 1 mL of CTAB. The tubes were shaken vigorously for 15 s and centrifuged at 12,000× *g* for 10 min at 4 °C. Following centrifugation, the aqueous phase was transferred into a new micro-centrifuge tube and centrifugation was repeated. The aqueous phase was collected and precipitated with one third volume of 8 M LiCl and incubated overnight at 4 °C, followed by centrifugation at 16,000 rpm for 30 min at 4 °C. The supernatant was removed, and the pellet was washed with 200 µL of 70% ETOH, followed by centrifugation at 13,000 rpm for 10 min at 4 °C. RNA quality was verified using a spectrophotometer (Nanodrop 3300, Wilmington, USA) and Bioanalyzer (Agilent 2100, Santa Clara, USA). Samples that showed higher RIN value were treated with DNAse Ambion DNA-free DNase Treatment and Removal kit (Cat #AM1906, California, CA, USA) following the manufacturing protocol (https://www.thermofisher.com/order/catalog/product/AM1906#/AM1906, last accessed on 01-07-2020). After DNAse treatment, RNA quality was analyzed again using Bioanalyzer. The mRNA of selected samples was fragmented and transformed to 100-bp short insert strand-specific cDNA libraries for sequencing on DNBseq PE100 from BGI (Europe). The raw reads from this library has been deposited in sequence read archive (SRA) with submission ID “PRJNA612415” (https://dataview.ncbi.nlm.nih.gov/object/PRJNA612415?reviewer=r5fg70lms7oandmb7aid7p0mib, last accessed on 10-04-2020).

### 5.4. Identification of Differentially Expressed Genes, Annotation and Gene Ontology

The clean reads were aligned to de novo assemblies of both hybrids included in this study using Kallisto [80,81]. Similarly, these reads were also aligned to the available draft reference genome of rye [82]. Differential gene expression analysis was performed in OmicsBox (Version 1.2.4) using the package EdgeR (Version 3.11) with false discovery rate (FDR) correction ≤ 0.05, *p* value ≤ 0.01, and fold change ≤ 2 or ≥2 [83]. EdgeR was also used to normalize the expected counts for relative expression and effective library size using the Trimmed Mean of M-values (TMM) normalization method. Genes with at least 1 count per million (CPM) were selected for further differential expression analysis. Differentially expressed genes (DEG) with FDR ≤ 0.05 and log fold change (logFC) of 2 were extracted for further analysis for gene ontology and enrichment analysis. GO mapping was performed against the Gene Ontology database implemented in OmicsBox (version 1.2.4) [84]. Sequences that shared similarities with known protein sequences in BLASTX searches with significant similarity (E < 1 × 10^−10^) were identified using the online tool InterProScan 5.0. The OmicBox program was used to assign Gene Ontology (GO) terms to the annotated sequences to predict the functions of the unique sequences with an e-value hit filter set to 1 × 10^−3^, annotation cutoff of 55, and evidence code set to 0.8 for the different categories as implemented in OmicsBox. Furthermore, KEGG analysis was used to identify potential pathways represented in the transcriptomes of DH372 and Helltop [85]. The OmicBox program was used to assign GO terms to the annotated sequences to predict the functions of the unique sequences and encoded translated proteins. The Venn diagrams were plotted using online tools Venny 2.0 http://bioinfogp.cnb.csic.es/tools/venny/.

### 5.5. In Silico Analysis of Differentially Expressed Genes

The orthologues of the differentially expressed genes belonging to cell wall modification and pectinesterase activity were identified in *T. aestivum*. All identified accession numbers of orthologs were analyzed manually and validated through BLAST search using the NCBI database (http://blast.ncbi.nlm.nih.gov/Blast.cgi, last accessed on 01-07-2020). The behaviors of these genes under various experimental conditions and in different plant parts were explored using the Genevestigator (www.genevestigator.com, last accessed on 31-08-2019). The Genevestigator is a manually curated and well-annotated database of expression profiling from 11 different plant species with more than 26,889 exclusive plant samples.

### 5.6. Cis Motif Identification and Occurrence of Transcription Factors in the Promoters of Identified Candidate Genes

The 1-kb upstream sequences from translational initiation codon of the identified genes belonging to the common and exclusive responding categories of DEGs were obtained. For this purpose, the orthologs of *T. aestivum* were used. The overrepresented *cis* motif consensus patterns were identified using the Multiple Expectation maximization for Motif Elicitation (MEME) analysis tool [86]. MEME was used to search for the best 5 *cis* motif based on E-value and consensus patterns of 6–50 base widths, only on the forward strand of the input sequences, and the distribution model used was the default Zero Or One Per Sequence (ZOOPS). A motif is a sequence pattern that occurs repeatedly in a group of DNA sequences. The motifs identified using MEME were analyzed through GOMO (Gene Ontology for Motifs) [87]. The purpose of GOMO is to identify possible roles (Gene Ontology terms) for DNA-binding motifs. GOMO takes a motif and determines which GO terms are associated with the (putative) target genes of the binding motif. GOMO was run using the “multiple species category”, which gave access to the plant database with a significant threshold at *q* < 0.01 and the number of scores shuffling rounds over 1000. In order to find the role of motifs associated to the transcription factor activity, a web-based tool “The Plant Promoter Analysis Navigator (PlantPAN 3.0; http://plantpan.itps.ncku.edu.tw/index.html) was utilized. This tool provides resources for detecting corresponding transcription factors associated to *cis* regulatory elements. Venn diagrams were plotted using online tool Venny 2.0 (http://bioinfogp.cnb.csic.es/tools/venny/, last accessed on 01-07-2020).

## Figures and Tables

**Figure 1 ijms-21-07418-f001:**
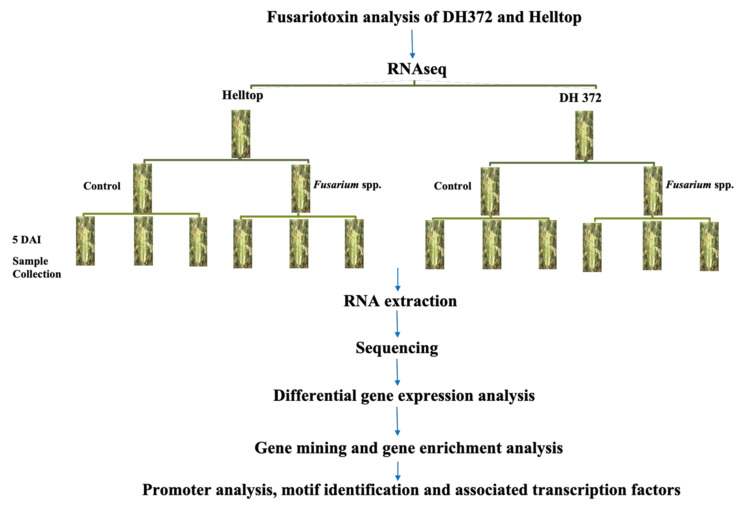
Graphical representation of the experimental setup and data analysis: this setup covers the fusariotoxin analysis, RNAseq experimental setup, RNA extraction, and subsequently next generation sequencing (NGS) data analysis; 5 DAI represents 5 days after inoculation.

**Figure 2 ijms-21-07418-f002:**
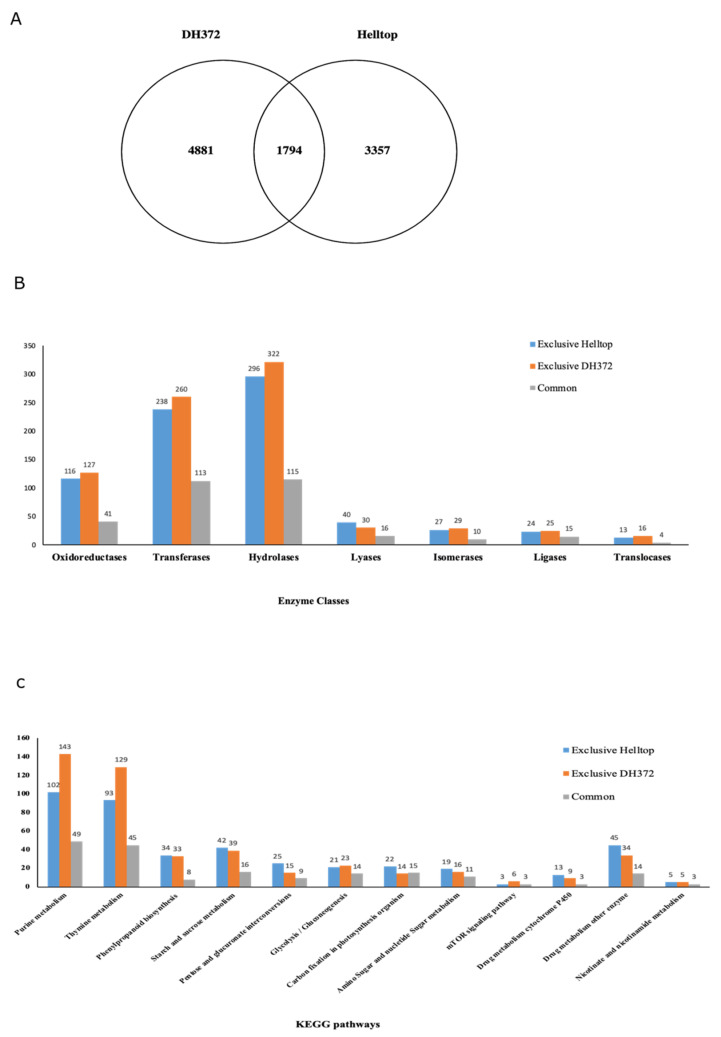
Distribution and annotations of differentially expressed genes (DEGs) of both hybrids after inoculation with *Fusarium* spp. through assigning enzyme classes and KEGG (Kyoto Encyclopedia of Genes and Genome) pathways: (**A**) the Venn diagram illustrates the number of differentially expressed genes in both hybrids and divides them into three categories. (**B**) The major classes of enzymes of DEGs associated to exclusive DH372, exclusive Helltop, and common category and (**C**) the top 12 most highly represented KEGG pathways belonging to each category are shown. Analysis was performed using the OmicsBox and the KEGG database.

**Figure 3 ijms-21-07418-f003:**
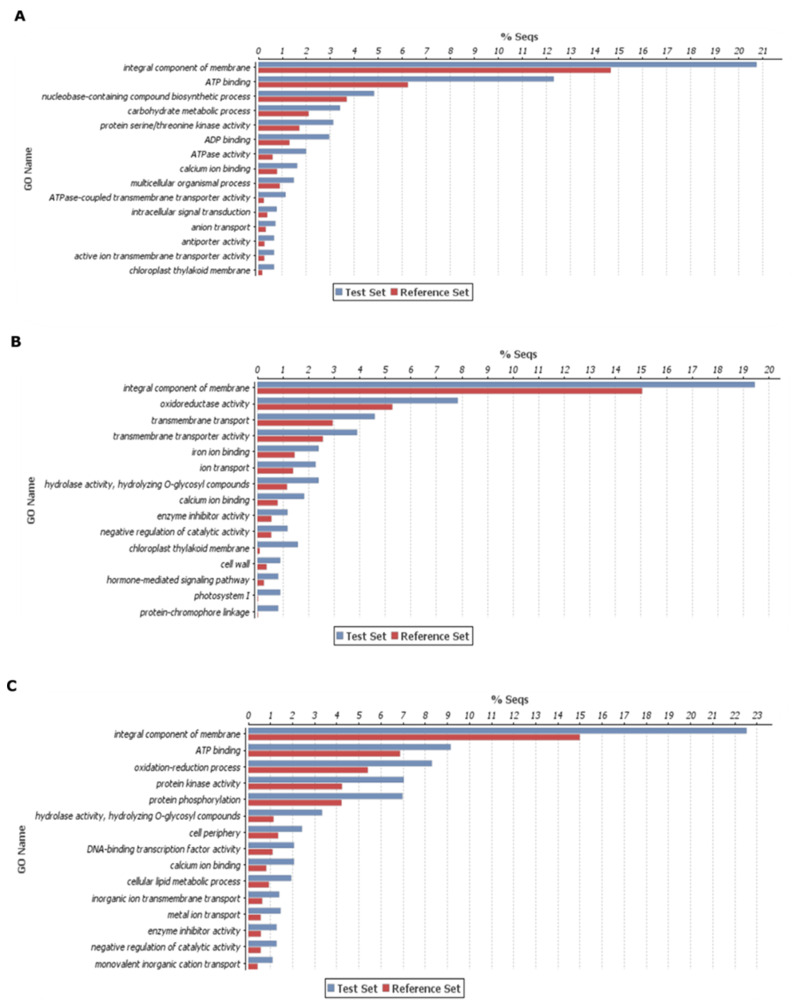
Gene enrichment analysis of differentially expressed genes (DEGs): this enrichment analysis was carried out using the upregulated genes after inoculation and belong to three different categories, i.e., (**A**) upregulated genes exclusive to DH372, (**B**) upregulated genes exclusive to Helltop, and (**C**) upregulated genes common in both hybrids. The top 15 significant gene-enriched gene ontology (GO) terms are shown. The test set represents the DEGs that were upregulated after inoculation, and the reference set is the total number of genes in rye known to be involved in that specific function.

**Figure 4 ijms-21-07418-f004:**
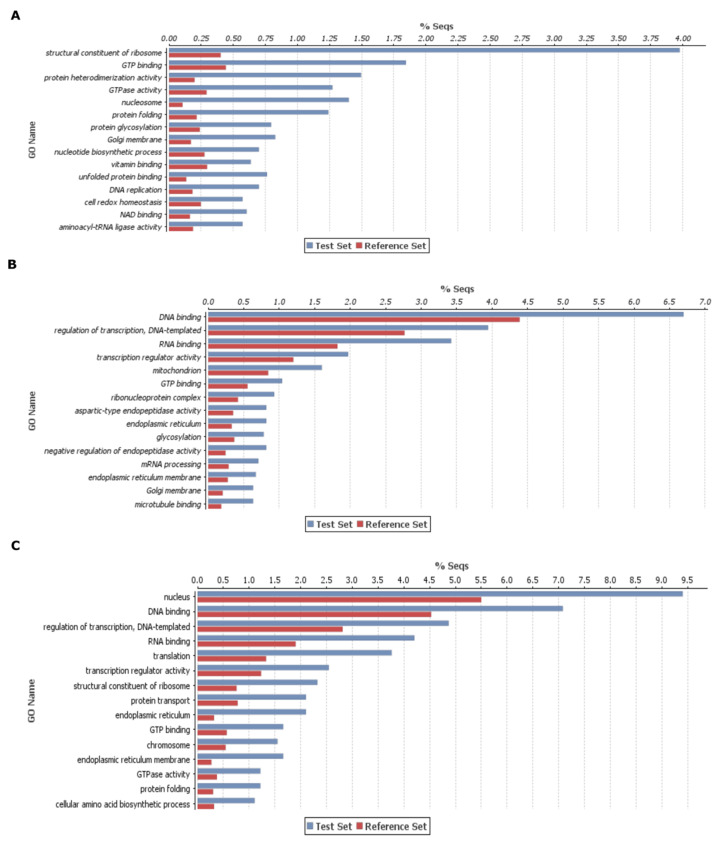
Gene enrichment analysis of differentially expressed genes (DEGs): this enrichment analysis was carried out using the downregulated genes after inoculation in both hybrids and belong to three different categories, i.e., (**A**) downregulated genes exclusive to DH372, (**B**) downregulated genes exclusive to Helltop, and (**C**) downregulated genes common in both hybrids. The top 15 significant gene-enriched gene ontology (GO) terms are shown. The test set represents the DEGs that were upregulated after inoculation, and the reference set is the total number of genes in rye known to be involved in that specific function.

**Figure 5 ijms-21-07418-f005:**
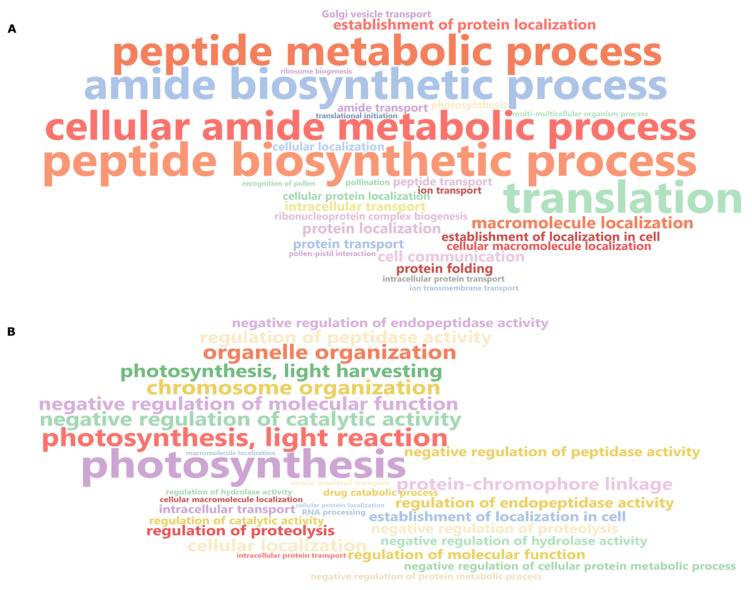
Gene set enrichment analysis (GSEA) of differentially expressed genes in both hybrids: (**A**) GSEA of DH372 and (**B**) GSEA of Helltop. The GSEA revealed that genes were recruited into different pathways in both hybrids after *Fusarium* spp. inoculation. The GSEA is presented in the form of a word cloud.

**Figure 6 ijms-21-07418-f006:**
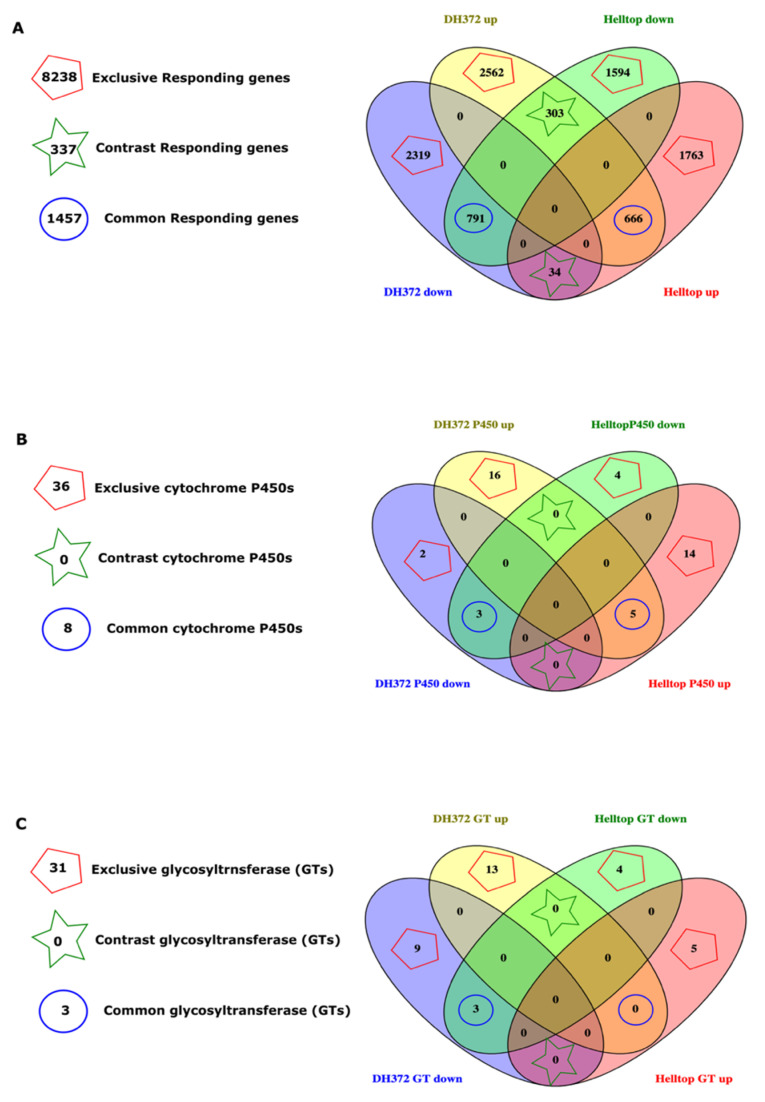
Dissection and mining of differentially expressed genes (DEGs): (**A**) the overall comparison of all DEGs was divided into three different categories known as exclusively responding genes, common responding genes, and contrasting responding genes categories. The exclusively responsive genes are the ones that were differentially expressed in one hybrid but absent in other. The common responding genes are the ones that exhibited similar differential expression pattern in both hybrids, whereas contrasting responding genes are the one that exhibited opposite response. (**B**) Comparison of cytochrome P450s in both hybrids and their division into three categories as mentioned above and (**C**) comparison of glycosyltransferases (GTs) in both hybrids and their division into the categories: the pentagons, stars, and circle encompass the total number of genes present in different comparisons. Venn diagrams were plotted using online tools Venny 2.0 (http://bioinfogp.cnb.csic.es/tools/venny/).

**Figure 7 ijms-21-07418-f007:**
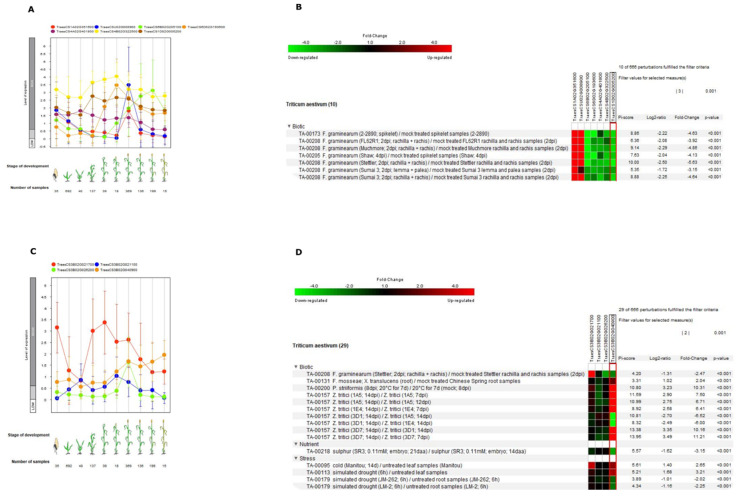
Gene expression patterns of wheat orthologs during different developmental stages and under various stresses: (**A**,**B**) these seven wheat orthologs belong to common responding genes, and (**C**,**D**) these four genes belong to exclusively responding genes in DH372. Developmental stage-specific expression patterns are shown in A and C. Our selected genes tend to show higher expression during anthesis. “HIGH,” “MEDIUM,” and “LOW” expressions were based on gene expression data found in GENEVESTIGATOR (http://www.genevestigator.com). Heat map of expression of selected genes in response to various stresses (**B**,**D**) using Genevestigator perturbation tool: the relative expression of the genes was represented in a log2 ratio, and significant changes in expression were filtered out based on *p* < 0.001. Expression of genes was strongly induced in response to biotic stresses including *Fusarium* spp.

**Table 1 ijms-21-07418-t001:** Field trial data and fusariotoxins analysis of DH372 and Helltop hybrids after inoculation with *Fusarium gramineraum* and *Fusarium culmorum*.

Name	Field Scoring *	NIV (µg kg^−1^)	DON (µg kg^−1^)	ZEN (µg kg^−1^)
Helltop	2	107	12,500	2474
DH372	1	26	1394	89

* Field scoring was done on a 0–10 scale, where 0 represents no disease and 10 represents very severe disease. The data indicate an average of 3 replications of each hybrid. The fusariotoxin of non-inoculated control plants has not been analyzed as the aim of this study was to detect the differential response of these two hybrids after artificial inoculation.

**Table 2 ijms-21-07418-t002:** The number of Differentially Expressed Genes (DEGs) in both hybrids with and without *Fusarium* spp. inoculation.

Treatment Comparison	Total DEGs	Downregulated Genes	Upregulated Genes
DH372 FHB vs. DH372 Control	6675	3144	3531
Helltop FHB vs. Helltop control	5151	2688	2463
Helltop FHB vs. DH372 FHB	4850	2623	2227
DH372 control vs. Helltop control	116	72	44

**Table 3 ijms-21-07418-t003:** List of cytochrome P450s and glycosyltransferases with their expression in both hybrids after pathogen infection (common responding category).

Gene	Description	Length	Tags	FC	logFC	logCPM	*p*-Value	FDR	Wheat Orthologs
DH372									
XLOC_114701	cytochrome P450 94B3-like	1593	UP	4	1.979	4.481	0.000	0.001	TraesCS1A02G351600
XLOC_996749	cytochrome P450 99A2-like	570	UP	4	1.852	2.031	0.001	0.002	TraesCSU02G008900
XLOC_939724	Cytochrome P450 71D7	3143	DOWN	−284	−8.148	9.684	0.000	0.000	TraesCS6B02G205100
XLOC_180117	cytochrome P450 86B1	1530	DOWN	−27	−4.735	6.473	0.000	0.000	TraesCS6D02G193600
XLOC_1029148	probable glycosyltransferase 7	1266	DOWN	−142	−7.1	9.4	0.000	0.000	TraesCS4A02G401800
XLOC_128937	glycosyltransferase	6226	DOWN	−6	−2.6	8.8	0.000	0.000	TraesCS4B02G322500
XLOC_1336988	glycosyltransferase-like At2g41451	4488	DOWN	−3	−1.4	6.5	0.000	0.000	TraesCS1D02G005200
Helltop									
XLOC_996749	cytochrome P450 99A2-like	570	UP	12	3.63	1.984	0.000	0.001	TraesCSU02G008900
XLOC_114701	cytochrome P450 94B3-like	1593	UP	3	1.567	2.959	0.001	0.008	TraesCS1A02G351600
XLOC_939724	Cytochrome P450 71D7	3143	DOWN	−473	−6.193	9.81	0.000	0.000	TraesCS6B02G205100
XLOC_180117	cytochrome P450 86B1	1530	DOWN	−3	−1.44	7.042	0.002	0.014	TraesCS6D02G193600
XLOC_1029148	probable glycosyltransferase 7	1266	DOWN	−56	−5.8	8.8	0.000	0.000	TraesCS4A02G401800
XLOC_128937	glycosyltransferase	6226	DOWN	−3	−1.7	8.6	0.000	0.001	TraesCS4B02G322500
XLOC_1336988	glycosyltransferase-like At2g41451	4488	DOWN	−2.4	−1.3	6.9	0.002	0.011	TraesCS1D02G005200

FC represents fold change (FC), FDR = false discovery rate.

**Table 4 ijms-21-07418-t004:** List of selected DEGs found in DH372 in the exclusive responding category.

Gene	Description	Length	Tags	FC	logFC	logCPM	*p*-Value	FDR	Wheat Orthologs
XLOC_1213553	Glycosyltransferase, HGA-like, putative, expressed	2889	UP	7	2.813	4.113	0.000	0.000	TraesCS3B02G021100
XLOC_165003	Cytochrome P450	2985	UP	5	2.248	5.423	0.000	0.000	TraesCS3B02G026200
XLOC_1267741	Glycosyltransferase, HGA-like, putative, expressed	2611	UP	2	1.132	5.569	0.000	0.001	TraesCS3B02G021700
XLOC_181573	Metal tolerance protein 7-like isoform X1	4859	DOWN	−4	−1.96	4.03	0.000	0.000	TraesCS3B02G040900

**Table 5 ijms-21-07418-t005:** Motifs found in the promoters of selected differentially expressed genes (DEGs) from the common responsive category in the Multiple Expectation maximization for Motif Elicitation (MEME) analysis: the top 5 common motifs identified by MEME in the promoters presented here with their E-value. The Gene Ontology for Motifs (GOMO) analysis provides information regarding the novelty of the motif and the probability that this motif is involved in certain cellular functions.

MEME Identified Motif Logo	Consensus Motif in Common Responding Genes	E-Value	Number of GO Terms Identified by GOMO	Role of Motif Identify by GOMO
1- 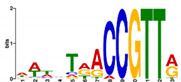	DADBTDACCGTTA	3 × 10^3^	1	MF transcription factor activity (MYB TFs)
2- 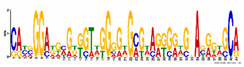	CAHVGGADGHGDKGTDGGGGDGMGDARKGRDGNABRGDGCA	1.6 × 10^−3^	05	MF transcription factor activity MF ATP binding CC mitochondrion CC nucleus BP protein import into nucleus
3- 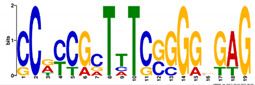	CCDYCGCTTTCSGGGNGAG	4.5 × 10^4^	2	CC chloroplast CC mitochondrion
4- 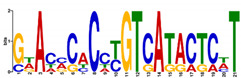	GDACVCACCYGTCATACTCHT	6.1 × 10^4^	0	
5- 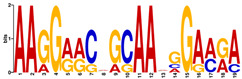	AAGGRRCNGCAANGGAMRA	1.4 × 10^5^	03	CC nucleus MF transcription factor activity CC plasma membrane

The E-value is an estimate of the expected number of motifs with the given log likelihood ratio (or higher) and with the same width and site count that one would find in a similarly sized set of random sequences (sequences where each position is independent and letters are chosen according to the background letter frequencies). The consensus sequences are presented in International Union of Pure and Applied Chemistry (IUPAC) nomenclature and constructed from each column in a motifs frequency matrix using the “50% rule”.

**Table 6 ijms-21-07418-t006:** Motifs found in the promoters of selected differentially expressed genes (DEGs) exclusively found in DH372 in the Multiple Expectation maximization for Motif Elicitation (MEME) analysis: the top 5 exclusive motifs identified by MEME in promoters presented here with their E-value. The Gene Ontology for Motifs (GOMO) analysis provides information regarding the novelty of the motif and the probability that this motif is involved in certain cellular functions.

MEME Identified Motif Logo	Consensus Motif in Common Responding Genes	E-Value	Number of GO Terms Identified by GOMO	Role of Motif Identify by GOMO
1- 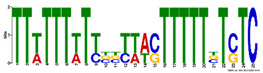	TTWTTTWTYBBYWWSTTTTTBTSTC	1.1 × 10^3^	05	MF transcription factor activityCC nucleusCC plasma membraneMF protein bindingBP regulation of transcription,
2- 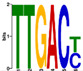	TTGACY	3.6 × 10^4^	01	MF transcription factor activity (WRKY TFs)
3- 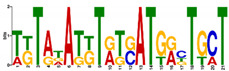	TKTADATKTRTSATGRBTGCT	6.9 × 10^4^	0	
4- 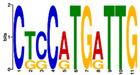	CTCCATGATTG	6.1 × 10^4^	0	
5- 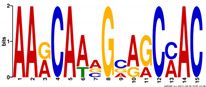	AARCAAVGVAGCMAC	1.2 × 10^5^	0	

The E-value is an estimate of the expected number of motifs with the given log likelihood ratio (or higher) and with the same width and site count that one would find in a similarly sized set of random sequences (sequences where each position is independent and letters are chosen according to the background letter frequencies). The consensus sequences are presented in the International Union of Pure and Applied Chemistry (IUPAC) nomenclature and constructed from each column in a motif’s frequency matrix using the “50% rule”.

**Table 7 ijms-21-07418-t007:** List of MYB family transcription factors with their expression in both hybrids after pathogen infection (common responding category).

Gene Name	Description	Length	Tags	FC	logFC	logCPM	*p*-Value	FDR	Wheat Orthologs
**DH372**									
**XLOC_130937**	myb family transcription factor EFM	2785	[UP]	9.66	3.27	3.59	0.000	0.000	TraesCS3B02G144100
**XLOC_1417787**	Myb family transcription factor APL	3656	[UP]	2.12	1.08	4.47	0.001	0.003	TraesCS7A02G381900
**XLOC_181060**	transcriptional activator Myb-like	2158	[DOWN]	−8.11	−3.02	3.38	0.000	0.000	TraesCS3B02G217100
**XLOC_1221063**	transcription factor MYB3R-1-like isoform X1	5586	[DOWN]	−9.48	−3.25	4.98	0.000	0.000	TraesCS3B02G180300
**XLOC_057451**	transcription factor MYB44-like	681	[DOWN]	−10.62	−3.41	3.06	0.000	0.000	TraesCS1A02G223600
**XLOC_1291741**	transcription factor MYB26-like	2467	[DOWN]	−23.54	−4.56	4.14	0.000	0.000	TraesCS2D02G234000
**Hellttop**									
**XLOC_130937**	Myb family transcription factor EFM	2785	[UP]	4.90	2.29	3.08	0.000	0.001	TraesCS3B02G144100
**XLOC_1417787**	Myb family transcription factor APL	3656	[UP]	2.43	1.28	5.08	0.008	0.039	TraesCS7A02G381900
**XLOC_181060**	transcriptional activator Myb-like	2158	[DOWN]	−3.98	−1.99	3.25	0.002	0.013	TraesCS3B02G217100
**XLOC_057451**	transcription factor MYB44-like	681	[DOWN]	−12.23	−3.61	2.23	0.000	0.000	TraesCS1A02G223600
**XLOC_1221063**	transcription factor MYB3R-1-like isoform X1	5586	[DOWN]	−32.43	−5.02	4.79	0.000	0.000	TraesCS3B02G180300
**XLOC_1291741**	transcription factor MYB26-like	2467	[DOWN]	−50.08	−5.65	2.59	0.000	0.000	TraesCS2D02G234000

**Table 8 ijms-21-07418-t008:** List of WRKY family transcription factors with their expression found in DH372 after pathogen infection (exclusive responding category).

Gene Name	Description	Length	Tags	FC	logFC	logCPM	*p*-Value	FDR	Wheat Orthologs
**XLOC_101792**	WRKY transcription factor WRKY24-like	3835	[UP]	13	3.71	4.39	0.000	0.000	TraesCS3B02G379200
**XLOC_1032358**	probable WRKY transcription factor 19	2504	[DOWN]	−5	−2.22	6.13	0.000	0.000	TraesCS7D02G234000
**XLOC_184351**	WRKY transcription factor 44-like	780	[DOWN]	−6	−2.59	4.19	0.000	0.000	TraesCS5D02G162000

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
