# Peer review of "A Comparative Transcriptome Analysis, Conserved Regulatory Elements and Associated Transcription Factors Related to Accumulation of Fusariotoxins in Grain of Rye (Secale cereale L.) Hybrids"

_ijms, 2020, doi:10.3390/ijms21197418_

Round 1
Reviewer 1 Report
The manuscript of Mahmood et al. describes the results of the transcriptome analysis of two rye hybrids that differ in mycotoxin accumulation after infection with F. graminearum and F. culmorum. FHB caused by these and other Fusaria is damaging cereal diseases in many grain-producing regions throughout the world, and grain contamination with fusariotoxins pose a serious threat to human health and create problems for animal husbandry. Therefore elucidation of the molecular mechanisms involved by plants to detoxify these mycotoxins is of considerable scientific interest and has great practical significance.
Using NGS technology, the authors identified metabolic processes and transcription factors preventing accumulation of mycotoxins in rye grain. The results provide novel information that is of interest to scientific community and may find implementation in breeding for the resistance to FHB. The methods used are properly described. The conclusions are sound.
However, the current manuscript version needs minor revision prior to publishing in IJMS. Additional information regarding verification of the NGS results would be quite desirable. The figures should be corrected and improved. English should be improved, and some sentences should be rephrased. See please comments listed below
Major comments
- I would recommend removing "content" from the title (the word seems redundant), and replacing "mycotoxins" with "fusariotoxins".
- At least for some of selected genes, transcriptome data should be verified by real-time PCR. If such data are available, it would be very useful to provide them and add the results of qPCR analysis.
- Figures are of poor quality. The captions are illegible, especially in Fig. 7. Fig.6A is referred in the text, but “A” is missed in the figure 6.
- In Tables 5 and 6, in several instances, there is no correspondence between the consensus sequence and the Logo plot. For example, Logo no. 4 in Table 5, the sequence begins with GDA, and in Logo, there is no D at position 2. The same sequence ends with CHT, and in Logo, there is no H. The same holds true for all sequences presented in Table 5. It therefore, remains unclear how these Logo plots were constructed.
Furthermore, all plots must be aligned relative to each other and given in the same scale.
Minor comments
The English language should be improved to avoid such clumsy word combinations as “DEGs …..were higher in the susceptible hybrid” (line 28), “Wheat orthologous” in Table 3 (Do you meant orthologs?), “Orthologous of all these…” (line 247) and others.
Please take into account that zearalenone is usually abbreviated as ZEN or ZEA.
Why HT and T2 were quantified if the pathogens used for inoculation do not produce them? The current version of the explanatory phrase on lines 122-123 confuses a reader. Please, rephrase, for instance, “…As expected, no HT and T2 that are not produced by the strains used in this study were found in the inoculated grain.” What about the content of the analyzed mycotoxins in grain of non-inoculated control plants because of a possible natural infection? I would suggest either add the control level in Table 1 or, alternatively, indicate in the text or under the table that in non-inoculated plants“….no toxins were detected” in or their level was“…below the detection limit”.
Consider editing a fragment on line 247 e.g., change to :"… all three those were also found to express differentially in ….."
Check and correct abbreviation of “species” in plural in Fig. 1, on line 44, 130 and throughout the text. It should be spp. (not italic !, full stop).
Check and correct misprints, especially especially large / small letters and italics / regular font (e.g. on lines 46, 132, 142, 195, 226, line 2 on page 12, line 128 and 133 on p.21) throughout the text.
Author Response
Reviewer 1
Comments and Suggestions for Authors
The manuscript of Mahmood et al. describes the results of the transcriptome analysis of two rye hybrids that differ in mycotoxin accumulation after infection with F. graminearum and F. culmorum. FHB caused by these and other Fusaria is damaging cereal diseases in many grain-producing regions throughout the world, and grain contamination with fusariotoxins pose a serious threat to human health and create problems for animal husbandry. Therefore elucidation of the molecular mechanisms involved by plants to detoxify these mycotoxins is of considerable scientific interest and has great practical significance.
Using NGS technology, the authors identified metabolic processes and transcription factors preventing accumulation of mycotoxins in rye grain. The results provide novel information that is of interest to scientific community and may find implementation in breeding for the resistance to FHB. The methods used are properly described. The conclusions are sound.
However, the current manuscript version needs minor revision prior to publishing in IJMS. Additional information regarding verification of the NGS results would be quite desirable. The figures should be corrected and improved. English should be improved, and some sentences should be rephrased. See please comments listed below
Response:
Thank you very much for highlighting the significance of study and the appreciation of our work.
Major comments
- Point 1: I would recommend removing "content" from the title (the word seems redundant), and replacing "mycotoxins" with "fusariotoxins".
Response 1:
This is a very nice suggestion and make the title more precise and disclose true reflection of study. The title has been changed to “A comparative transcriptome analysis, conserved regulatory elements and associated transcription factors related to accumulation of fusariotoxins in grain of rye (Secale cereale L.) hybrids” as suggested. Please see line 2-5.
- Point 2: At least for some of selected genes, transcriptome data should be verified by real-time PCR. If such data are available, it would be very useful to provide them and add the results of qPCR analysis
Response 2:.
Thank you for raising this issue. Initially, we also considered performing qPCR studies to ‘re-validate’ some of our gene-expression findings but there is little evidence that qPCR analyses from the same samples will add any extra value to our data so we decided to avoid those experiments. Previous studies have shown extremely close correlations between qPCR and RNAseq data [1-4]. RNAseq is generally more accurate than q-PCR. Moreover, q-PCR is advised in case of very few or no biological replicates but in our case, we have three independent biological replicates for each treatment. Hence, we disagree with the need to validate RNAseq with RT-qPCR here. Ideally, we would re-validate our findings (potentially by qPCR) in a separate cohort of samples, but due to the difficulty in accessing these samples, those experiments are not possible at this time.
- Point 3: Figures are of poor quality. The captions are illegible, especially in Fig. 7. Fig.6A is referred in the text, but “A” is missed in the figure 6.
Response 3:
In this revised version, we had provided new figures with higher and better resolution. We are also providing the figure in a separate file in PNG format and if needed please use them. Fig 6 has been re-drawn see page 13. The captions and legends are inserted at appropriate places. Overall, now the figures are more self-explanatory in nature with better resolution.
- Point 4: In Tables 5 and 6, in several instances, there is no correspondence between the consensus sequence and the Logo plot. For example, Logo no. 4 in Table 5, the sequence begins with GDA, and in Logo, there is no D at position 2. The same sequence ends with CHT, and in Logo, there is no H. The same holds true for all sequences presented in Table 5. It therefore, remains unclear how these Logo plots were constructed. Furthermore, all plots must be aligned relative to each other and given in the same scale
Response 4:
Thank you very much for highlighting it. The consensus sequences are presented in IUPAC nomenclature and constructed from each column in a motif's frequency matrix using the "50% rule". In IUPAC nomenclature, D represent the nucleotide A, G and T, and H represent A, C and T. For example, logo no.4 in Table 5 has consensus sequence “GDACVCACCYGTCATACTCHT” where G has frequency more than 50% and hence we keep G. Position two has “D” because frequency of A, G and T is equal in the promoters of studied genes and hence we used D. Position 3 has A due to frequency more than 50%. Position 5 has V because frequency of A, C and G equal. Position 10 has Y due to equal frequency of C and T. We had used this principle for all other consensus motifs presented in table 5 and table 6. This is a standard way of presenting the motif in an article. We had provided the explanation at the end of each table now. See page 17 and 18 at the end of Table 5 and 6.
We have tried to align the plots. However, the logo contain different number of sequences hence it’s not a good idea in the same scale.
Minor comments
Point 5: The English language should be improved to avoid such clumsy word combinations as “DEGs …..were higher in the susceptible hybrid” (line 28), “Wheat orthologous” in Table 3 (Do you meant orthologs?), “Orthologous of all these…” (line 247) and others.
Response 5:
We had improved the English language and made suggested changes throughout manuscript. See line 28, line 332 and tables 3, 4, 7 and 8.
Point 6: Please take into account that zearalenone is usually abbreviated as ZEN or ZEA.
Response 6:
We had modified it throughout the MS and abbreviated as ZEN. For instance, see line 24, 70, 135, 138 and table 1.
Point 7: Why HT and T2 were quantified if the pathogens used for inoculation do not produce them? The current version of the explanatory phrase on lines 122-123 confuses a reader. Please, rephrase, for instance, “…As expected, no HT and T2 that are not produced by the strains used in this study were found in the inoculated grain.” What about the content of the analyzed mycotoxins in grain of non-inoculated control plants because of a possible natural infection? I would suggest either add the control level in Table 1 or, alternatively, indicate in the text or under the table that in non-inoculated plants“….no toxins were detected” in or their level was“…below the detection limit”.
Response 7:
The fusariotoxin accumulation (DON, NIV, ZEN, T-2, and HT-2) in the seeds of inoculated hybrids were quantified in an LC-MS/MS system consisting of an Agilent 1260 infinity chromatographic system (Santa Clara, CA, USA) coupled to an AB Sciex 3200 triple quadrupole trap mass spectrometer (QTRAP/MS) (AB Sciex, Framingham, USA). We have optimized this for all fusariotoxins, hence we achieved quantification of HT and T2 automatically. Moreover, this provide the validation of fusariotoxins analysis and indicate that using this method we can correctly distinguish level of various fusariotoxins.
We did not analyze the fusariotoxin in the grains of non-inoculated control plants. The aim of our study is to provide comparison of two hybrids after inoculation. Now we had indicated that we did not analyzed the fusariotoxin in grains of control plants. See line 144
Point 8: Consider editing a fragment on line 247 e.g., change to :"… all three those were also found to express differentially in ….."
Response 8:
We have restructure the complete paragraph and changed as “Interestingly, four genes in exclusive responding category of DH372 are also identified in wheat and involved in FHB resistance [41]. Among them, three were identified as P450s and glycosyltransferase. These genes were named as “XLOC_1213553”, “XLOC_165003” and “XLOC_1267741” with upregulation at 7, 5 and 2 fold changes, respectively” see line 254-258
Point 9: Check and correct abbreviation of “species” in plural in Fig. 1, on line 44, 130 and throughout the text. It should be spp. (not italic !, full stop).
Response 9:
This has been corrected throughout manuscript. Please see Fig 1, line 50, 120, 140, 162, 175 and so on.
Point 10: Check and correct misprints, especially especially large / small letters and italics / regular font (e.g. on lines 46, 132, 142, 195, 226, line 2 on page 12, line 128 and 133 on p.21) throughout the text.
Response 10:
The suggested changes have been made, please see Line 51, 165, 175, 265 and 315. Similarly, page 12 line 129 and 134 on page 21.
We prepared the revision based on the issues raised by reviewer, change title of MS to reflect objective of study as reviewer suggested and abide by the format requirements mentioned in the guidelines. We double check the MS and tried to improve the expression and grammar throughout the manuscript in this second revision. We grateful to the reviewer for the constructive feedback which has improved manuscript substantially.
References
- Griffith, M.; Griffith, O.L.; Mwenifumbo, J.; Goya, R.; Morrissy, A.S.; Morin, R.D.; Corbett, R.; Tang, M.J.; Hou, Y.-C.; Pugh, T.J. Alternative expression analysis by RNA sequencing. Nature methods 2010, 7, 843.
- Wu, A.R.; Neff, N.F.; Kalisky, T.; Dalerba, P.; Treutlein, B.; Rothenberg, M.E.; Mburu, F.M.; Mantalas, G.L.; Sim, S.; Clarke, M.F. Quantitative assessment of single-cell RNA-sequencing methods. Nature methods 2014, 11, 41.
- Shi, Y.; He, M. Differential gene expression identified by RNA-Seq and qPCR in two sizes of pearl oyster (Pinctada fucata). Gene 2014, 538, 313-322.
- Hughes, T.R. 'Validation'in genome-scale research. Journal of biology 2009, 8, 3.
Reviewer 2 Report
All the figures are quite small in size and of very low resolution. The authors must provide all the figures in a readable size and of high quality. No information are clear and therefore cannot be reviewed.
Author Response
Reviewer 2
Comments and Suggestions for Authors
All the figures are quite small in size and of very low resolution. The authors must provide all the figures in a readable size and of high quality. No information are clear and therefore cannot be reviewed.
Response:
All the figures have been reproduced and inserted at appropriate place in the text. The figure are also provided in separate file as PNG format with better resolution. The captions and legends are inserted at appropriate places. Overall, now the figures are self-explanatory in nature.
We double check the MS and tried to improve the expression and grammar throughout the manuscript in this second revision. Overall, we believed that manuscript is in a good shape to meet Journal's standards for clarity and accuracy.Reviewer 3 Report
The paper describes a study on the transcriptomic analysis of the two rye hybrids under Fusarium infection conditions to reveal the candidate genes potentially involved in the detoxification process of the main mycotoxins produced and accumulated in the grain. The article reports some new data but it is not very well-written and requires extensive editing before it can be processed further. English should also be improved in terms of grammar and style. Major concerns and suggestions were outlined below.
The Introduction is chaotic and lacks focus. There is a lot of information available concerning FHB research, including epidemics, pathogen populations, mycotoxin synthesis and plant resistance types, so here it should be given in the ordered manner. I would suggest to elaborate the aim of using mixed infection and analyzing the metabolites that are known not to be produced by the strains of the species used. The chemotypes of the strains used for inoculation is also not known. Furthermore, it is interesting that the Authors selected two hybrids that did not differ in FHB severity. This choice should also be better justified, as the differences observed for individual DEGs can come from the genotype specificity and not be directly linked to the specific type of FHB resistance, as the Authors have claimed. I am not convinced by the statement given in lines 106-109 as the study was not designed for revealing the “genetic architecture” of the trait. Please, re-phrase it with caution.
Results: T-2 and HT-2 toxin analysis is distracting as both species used for inoculation do not produce these mycotoxins. Consider deleting this part of results and focus on the transcriptomic analyses. From the description it is not clear how the samples for the RNAseq were obtained and prepared. I would suggest to add some explanation in the beginning of the paragraph. Alternatively, Figure 1 could be moved there as it is informative and helps to follow the story. The number of common DEGs seems to be low, regardless of the gene class annotation (Figure 2), particularly that the hybrids share a common genetic background (to some extent at least). I think this issue should be discussed in more detail. Why only top 12 KEGG categories were chosen for further analysis?
Discussion is mostly speculative as no experimental confirmation was done to prove the involvement of the DEGs identified, cis elements and TFs predicted to be involved in the resistance processes. I would recommend the Authors, to re-write it carefully and avoid statements that might be too strong.
Materials and Methods: I would recommend to describe the inoculation procedure in more detail, explaining why such combination of strains was chosen and how exactly it was used. Also, defining their chemotypes would be beneficial for the reliability of the results. This section is detailed enough to follow the procedures and repeat the experiments. I do not have major questions to it.
Author Response
Point by point response
Reviewer 3
The paper describes a study on the transcriptomic analysis of the two rye hybrids under Fusarium infection conditions to reveal the candidate genes potentially involved in the detoxification process of the main mycotoxins produced and accumulated in the grain. The article reports some new data but it is not very well-written and requires extensive editing before it can be processed further. English should also be improved in terms of grammar and style. Major concerns and suggestions were outlined below.
Response:
Thank you very much for acknowledging the presentation of new data in the Manuscript. We have done intensive editing and tried to improve the language in terms of grammar and style. Please find below our point by point response to the issue raised. Also, thanks for you many specific writing suggestions that has been a big improvement of the overall readability of the manuscript.
Major comments
- Points regarding introduction:
- The Introduction is chaotic and lacks focus. There is a lot of information available concerning FHB research, including epidemics, pathogen populations, mycotoxin synthesis and plant resistance types, so here it should be given in the ordered manner. I would suggest to elaborate the aim of using mixed infection and analyzing the metabolites that are known not to be produced by the strains of the species used. The chemotypes of the strains used for inoculation is also not known.
Response 1:
This is a fair comment. We have done intensive editing in the introduction section. In the revised MS, we have now presented the information concerning FHB research in an ordered manner in the first paragraph without compromising the aim of our study. Please see detail in the first paragraph of introduction section line 45-72
- Furthermore, it is interesting that the Authors selected two hybrids that did not differ in FHB severity. This choice should also be better justified, as the differences observed for individual DEGs can come from the genotype specificity and not be directly linked to the specific type of FHB resistance, as the Authors have claimed.
Response 2:
We have selected two hybrids for transcriptome analysis because these two differ in the accumulation of mycotoxin upon pathogen infection. It is true that there is no big difference visually in the fusarium infection but there is a huge difference in the toxin content. Fusarium resistance can be divided into 5 different types. Types 1 is resistance to initial infection and 2 is the resistance for spreading of the infection. Type 3 is resistance to kernel infection. Type 4 is tolerance to infection and type 5 is resistance to mycotoxins production (see line 56-60). A line can possess different degrees of resistance to each of these types and therefore, it is normal to observe resistance variation in one type but not necessarily in other types of fusarium resistance.
The individual DEGs cannot be due to genotype specificity alone as these were not identified when comparison was made between the untreated controls of these two hybrids. What we found is differences in how the hybrids respond to toxin accumulation.
- I am not convinced by the statement given in lines 106-109 as the study was not designed for revealing the “genetic architecture” of the trait. Please, re-phrase it with caution
Response 3:
This statement has been replaced with “The differential expression of disease responsive genes were profiled and the molecular pathways involved were analyzed. It is expected that this work will serve as valuable scientific resource for better understanding of specific type of FHB resistance (Type V) in rye. See line 106-107, 114-115.
- Points regarding results:
- T-2 and HT-2 toxin analysis is distracting as both species used for inoculation do not produce these mycotoxins. Consider deleting this part of results and focus on the transcriptomic analyses.
Response 1:.
As the species used for inoculation do not produce T-2 and HT-2 toxin, we did not observe difference in these toxins between two hybrids. Hence in order to avoid distraction, we had deleted information regarding T-2 and HT-2 (see line 121-122 and Table 1)
- From the description it is not clear how the samples for the RNAseq were obtained and prepared. I would suggest to add some explanation in the beginning of the paragraph. Alternatively, Figure 1 could be moved there as it is informative and helps to follow the story.
Response 2:
We provided the detailed description explaining how the samples for RNAseq was prepared and obtained. Please see the material and method section page 24, line 168-170. Moreover, we had moved the figure 1 before the results description of transcriptome analysis see figure 1 on page 4.
- The number of common DEGs seems to be low, regardless of the gene class annotation (Figure 2), particularly that the hybrids share a common genetic background (to some extent at least). I think this issue should be discussed in more detail. Why only top 12 KEGG categories were chosen for further analysis?
Response 3:
The low number of common DEGs upon pathogen infection represents the differential response of the hybrids. This was also evident in the fusariotoxin analysis where one hybrid accumulated higher amount of toxin compared to other. It is true that these two hybrids have the same genetic components except the restoring component, but as they differed substantially in their resistance to build up of fusariotoxins it is not surprising that the detected DEGs will differ between them. Some of these different DEGs will be causal for the difference in fusariotoxins and some of them might be responding to the fusariotoxins levels and therefore differ between the hybrids. The study conducted by Biselli et al., 2018 regarding the transcriptome profiles of two near isogenic lines of bread wheat after Fusarium infection revealed that 33% DEGs (8201 out of 24755) of susceptible line were common between both genotypes. In our case, 35.5% DEGs (1794 out of 5051) of Helltop (susceptible hybrid) were common in both hybrids upon pathogen infection. Hence low number of common DEGs between two genotype is not unusual after Fusarium infection.
Due to high number of DEGs and their distribution into various KEGG pathways, this is the common practice in transcriptome studies to select most highly represented KEGG pathways. We have selected top 12 most highly represented KEGG pathways and there is no specific reason for the number.
- Points regarding discussion: Discussion is mostly speculative as no experimental confirmation was done to prove the involvement of the DEGs identified, cis elements and TFs predicted to be involved in the resistance processes. I would recommend the Authors, to re-write it carefully and avoid statements that might be too strong
Response:
Thank you for pointing towards this aspect. We have obtained a large number of DEGs and in the process of gene mining and identification of candidate genes associated to detoxification of fusariotoxins, we conducted various in silico analysis. However, this is true that we do not have experimental confirmation about the involvement of identified cis elements and TFs. Hence, we have done thorough editing in this section and soften the statements that seems too strong. We have been consistent to use words such as assumed, and speculate where we do not have experimental evidence of our findings See discussion section on page 20-22.
- Points regarding material and methods:
Materials and Methods: I would recommend to describe the inoculation procedure in more detail, explaining why such combination of strains was chosen and how exactly it was used. Also, defining their chemotypes would be beneficial for the reliability of the results. This section is detailed enough to follow the procedures and repeat the experiments. I do not have any other major questions to it.
Response:
Thank you very much for highlighting this point. Both species (F. culmorum and F. graminearum) used in this study are characterized as DON producers. In previous studies it has been found that selected strains of these species are common in Denmark (Nielsen et al., 2010). These were the main reason for the selection of these species. This has been added in the material and method section see line 142-143 on page 23.
The method use for inoculation and spray has been a standard for screening of wheat cultivars susceptibility to Fusarium head blight for many years (Etzerodt et al, 2016). The spikes were sprayed directly with a Fusarium spp. macro-conidial suspension (2 × 106 spores/ml) three times during anthesis using the knapsack sprayer See line 144-146 on page 23.
References
- Biselli, Chiara, et al. "Comparative transcriptome profiles of near-isogenic hexaploid wheat lines differing for effective alleles at the 2DL FHB resistance QTL." Frontiers in plant science9 (2018): 37.
- Nielsen, Linda Kærgaard, et al. "Fusarium head blight of cereals in Denmark: species complex and related mycotoxins." Phytopathology8 (2011): 960-969.
- Etzerodt, T.; Gislum, R.; Laursen, B.B.; Heinrichson, K.; Gregersen, P.L.; Jørgensen, L.N.; Fomsgaard, I.S. Correlation of deoxynivalenol accumulation in Fusarium-infected winter and spring wheat cultivars with secondary metabolites at different growth stages. Journal of agricultural and food chemistry (2016) 64, 4545-4555.
Round 2
Reviewer 2 Report
no improvement was done.
Author Response
We have done intensive editing in the structure of MS. We have improved the language in terms of grammar and style. The changes we made in this revision has been a big improvement of the overall readability of the manuscript. Also we appreciate the anonymous reviewer for many specific writing suggestions that have been very useful.
Reviewer 3 Report
The Authors addressed all of the crucial issues raised in the review and the revised manuscript reads much better now. The improvements are satisfactory, I do not have further complaints.